Cancer histology in metastatic lymph node predicts prognosis in patients with node-positive stage IV colorectal cancer

http://orcid.org/0000-0003-1301-6434 Yokoyama Shozo yokoyama@wakayama-med.ac.jp
Watanabe Takashi
Matsumura Shuichi
Tamiya Masato
Nagano Shotaro
Hori Yuya
Department of Surgery, National Hospital Organization Minami Wakayama Medical Center , Tanabe, Wakayama , Japan
Jurisic Vladimir
Electronic publication date: 2024 Jul 9
Publication date: 2024
Volume: 12
Electronic Location ID: e17702
Received 2024 Apr 29; Accepted 2024 Jun 17
Copyright: © 2024 Yokoyama et al.
Copyright year: 2024
Copyright holder: Yokoyama et al.
License: This is an open access article distributed under the terms of the Creative Commons Attribution License, which permits unrestricted use, distribution, reproduction and adaptation in any medium and for any purpose provided that it is properly attributed. For attribution, the original author(s), title, publication source (PeerJ) and either DOI or URL of the article must be cited.
License URL: https://creativecommons.org/licenses/by/4.0/

Keywords: Lymph node metastasis, Histology, Metastatic colorectal cancer

Funding: The authors received no funding for this work.

==============================
Background

Appropriate prognostic indicators are required for patients with stage IV colorectal cancer (CRC). Lymph node metastasis mainly involves four histological types of CRC. Some metastatic lymph nodes (mLNs) showing cribriform carcinoma are associated with distant metastasis in patients with node-positive CRC and are correlated with recurrence and survival in stage III disease. However, the significance of mLN histology in the prognosis of patients with node-positive stage IV disease remains unclear.

Methods

We enrolled 449 consecutive patients with CRC who underwent primary tumor resection with lymph node dissection between January 2011 and November 2018. This study included 88 patients with node-positive stage IV CRC and synchronous or metachronous distant metastases. We retrospectively investigated the association between cancer histology in the mLNs based on our classification and cancer-specific survival (CSS) in patients with node-positive stage IV CRC.

Results

Kaplan-Meier analysis showed that CSS was better in patients with CRC and all the mLNs showing tubular-type carcinoma. In contrast, patients with at least some mLNs showing poorly differentiated-type carcinoma had poor prognosis. Multivariate analysis showed that “all mLNs showing tubular-type carcinoma” was an independent good prognostic factor for CSS in patients with node-positive stage IV CRC. In addition, “at least some mLNs showing poorly differentiated-type carcinoma” was an independent poor prognostic factor for CSS in patients with node-positive stage IV disease.

Conclusions

The histological type of the mLN may indicate a better or poor prognosis for patients with stage IV CRC.

Introduction

The prognosis of stage IV colorectal cancer (CRC) has improved with the development of anticancer drugs. Treatment guidelines for stage IV CRC have been published based on the results of clinical trials (Cervantes et al., 2023; Morris et al., 2023). Advances in treatment have resulted in prolonged cancer-specific survival (CSS). Optimal selection of a better or poor prognosis group will allow for more effective treatments to be administered. Studies targeting various predictive factors (Cheng et al., 2022; Kawamura et al., 2022; Miyakawa et al., 2022; Nicolazzo et al., 2023; Sayagues et al., 2023; Shibutani et al., 2023; Su et al., 2022; Wang et al., 2023; Wei et al., 2023; Wu et al., 2022; Ziranu et al., 2023; Zozaya et al., 2023) have reported the prognosis of patients with stage IV CRC. However, these predictive factors have not been used as clinical criteria. Therefore, it is important to explore more accurate predictors to select better or poor prognostic cases of stage IV CRC.

Cancer histology in CRC lymph node metastasis (LNM) includes four histological types (Yokoyama et al., 2023), and “some metastatic lymph nodes (mLNs) showing cribriform carcinoma” is associated with distant metastasis in patients with node-positive CRC and is correlated with recurrence and survival in stage III disease (Yokoyama et al., 2021). The significance of CRC histology in mLNs for patients with stage IV CRC remains unclear. This study explored the significance of cancer histology in mLNs for predicting prognosis in patients with node-positive stage IV CRC.

Materials and Methods

Patients

This study retrospectively enrolled 449 consecutive patients with CRC who underwent colectomy, anterior resection, Hartmann’s operation, or abdominoperineal resection with lymphadenectomy between January 2011 and November 2018 at the National Hospital Organization Minami Wakayama Medical Center (Tanabe, Japan). The follow-up period was 5 years. This study was approved by the ethics committee of the National Hospital Organization Minami Wakayama Medical Center (#2023-11). All methods were performed in compliance with the Declaration of Helsinki, the Guidelines for Ethical Principles for Medical Research Involving Human Subjects, and the Ethics Guidelines of the National Hospital Organization Minami Wakayama Medical Center. Informed consent was obtained in the form of opt-out on the web page of the National Hospital Organization Minami Wakayama Medical Center.

Histological analysis

Colorectal and lymph node tissue sections were stained using hematoxylin & eosin stain. The surgical specimens were fixed in a 10% buffered formaldehyde solution, washed under running tap water, dehydrated, and cleared in xylene. Paraffin embedding was performed. Sections of 5 μm were cut and placed on glass slides. The specimens were deparaffinized in xylene and rehydrated in a degressive series of ethanol solutions. The specimens were incubated in hematoxylin solution for 10 min at room temperature (RT, 15–30 °C), rinsed in distilled water, and immersed in acid alcohol for 1 min. The slides were then washed under running tap water for 10 min at 40 °C, incubated in eosin solution for 4 min and rinsed in distilled water for a few seconds. Subsequently, the slides were immersed in 100% ethanol for 2 min at RT and then dehydrated in xylene for 2 min at RT and mounted on cover slides. The slides were then examined under a microscope. Data were collected as previously described in Yokoyama et al. (2023). Specifically, all slides were blindly reviewed twice by three individuals (SY, TW, and SM). Discrepancies regarding the specimens’ observations were discussed to reach a consensus. All images were acquired using an Olympus CX33 (Olympus, Tokyo, Japan) with an NY1S adaptor (Micronet, Torrence, CA, USA), EOS X9, and the EOS utility software program (Canon, Tokyo, Japan). As previously reported (Yokoyama et al., 2023), the histological classification of mLNs was performed based on the shape of the hollow cavities and cellular polarity. “Tubular-type” cells have an elongated hollow cavity and cancer cell polarity (Fig. 1A). “Cribriform-type” has small rounded cavities and no cancer cell polarity (Fig. 1B). “Poorly differentiated-type” has almost no hollow cavities and no cancer cell polarity (Fig. 1C). “Mucinous-type” contains mucin in the cavity (Fig. 1D).

Figure 1 Histologies in metastatic lymph node.

(A) Tubular-type carcinoma; (B) cribriform-type carcinoma; (C) poorly differentiated-type carcinoma; (D) mucinous-type carcinoma.

Statistical analyses

Pearson’s chi-square test was used to compare clinicopathological factors. The Kaplan–Meier method was used to estimate postoperative survival, and the log-rank test was used to determine statistical significance. The Cox proportional hazards model was used to assess the risk ratio with simultaneous contributions from several covariates. A p < 0.05 was considered statistically significant. Calculations for the Pearson’s chi-square test, Kaplan–Meier method, log-rank test, and Cox proportional hazards model were performed using the JMP Pro software application (version 14.1.0; SAS Institute, Cary, NC, USA). Schoenfeld residuals were tested to examine the validity of the proportional hazard assumption using the STATA software application (version 13.1.0; StataCorp, College Station, TX, USA).

Results

Patient characteristics

Of 449 patients with CRC, 88 (19.6%) patients with node-positive stage IV CRC who had synchronous (n = 43) or metachronous (n = 45) distant metastases were included in this study. The mean age of the patients was 71 years (range: 34–90 years). This study included 48 males and 40 females. The resected tumors were located in the colon (n = 51) or rectum (n = 37). Regarding T stages, 57, 27, and four patients had T3, T4a, and T4b stage tumor, respectively. Patients were divided into four groups based on the classification of histology in mLNs as previously reported: tubular-type group (n = 16) with all mLNs showing tubular carcinoma (Fig. 1A), cribriform-type group (n = 56) with some mLNs showing cribriform carcinoma (Fig. 1B), poorly differentiated-type group (n = 15) with some mLNs showing poorly differentiated carcinoma (Fig. 1C), and mucinous-type case (n = 1) with some mLNs showing mucinous carcinoma (Fig. 1D). Patients with stage III tumor received 5-fluorouracil-based adjuvant chemotherapy before metachronous relapse. Of the 88 patients with stage IV tumors, 22 (25%) patients underwent metastasectomy, including 12 liver resections (six patients with tubular-type and six patients with cribriform-type), six lung resections (two patients with tubular-type, three patients with cribriform-type, and one patient with mucinous-type), three liver and lung resections (one patient with tubular-type and two patients with cribriform-type), one lung resection and splenectomy (one patient with poorly differentiated-type). Two patients experienced local recurrence with distant metastases. The rate of local recurrence was 2.3%. Distant metastatic organs (the number of patients) were liver (30); liver and lung (15); peritoneum (five); lung (four); lung and distant lymph node (four); distant lymph node (three); bone (three); liver and peritoneum (three); liver and distant lymph node (three); lung and peritoneum (three); liver, lung and peritoneum (three); liver, lung and distant lymph node (three); lung and bone (two); lung, distant lymph node and bone (two); peritoneum and spleen (one); peritoneum and distant lymph node (one); liver, peritoneum and distant lymph node (one); lung, peritoneum and distant lymph node (one); and liver, lung, distant lymph node and kidney (one).

Histological types in mLN and clinicopathological factors in patients with node-positive stage IV CRC

The clinicopathological factors were compared among patients with different histological types in mLNs. The cases of poorly differentiated adenocarcinoma in the primary tumor were significantly more, with poorly differentiated carcinoma in the mLN. Cribriform-type carcinomas in mLNs was significantly associated with lower lymphatic permeation (Table 1).

Table 1 Histological types in metastatic lymph node and clinicopathological factors.

Variables	Histological types in metastatic lymph node	
Tubular-type
n = 16	Cribriform-type
n = 56	Poorly differentiated-type
n = 15	p-value	
Gender (male/female)	10/6	28/28	9/6	0.5959	
Age (>70 vs. ≦70)	9/7	32/24	11/4	0.4162	
Tumor site (rectum/colon)	10/6	24/32	3/12	0.0905	
Primary tumor differentiation (por/others)	0/16	1/55	6/9	<0.0001	
Tumor depth (T4 /T3)	2/14	21/35	8/7	0.0917	
Lymphatic permeation (presence/absence)	15/1	47/9	14/1	0.0465	
Venous permeation (presence/absence)	8/8	36/20	11/4	0.3130	
Number of mLNs (>3/≦3)	10/6	28/28	11/4	0.2422	
Distant metastasis (synchronous/metachronous)	9/7	26/30	7/8	0.6690	
PS ECOG (Grade 0/1/2/3/4)	16/0/0/0/0	56/0/0/0/0	15/0/0/0/0	–	
5-year follow-up rate (%)	100	100	100	–	
Note:

Tubular-type in mLNs, all mLNs showing tubular carcinoma; Cribriform type in mLNs, some mLNs showing cribriform carcinoma; Poorly differentiated-type in mLNs, some mLNs showing poorly differentiated carcinoma; PS, performance status; EOCG, Eastern Cooperative Oncology Group.

Comparison of CSS times among histological types of mLN in patients with node-positive stage IV CRC

To assess the impact of the histological types of mLN on CSS in patients with node-positive stage IV CRC, the tubular, cribriform, or poorly differentiated-types were compared. All patients were followed up for 5 years. There were no missing data or outliers. No adjustments were made to the Kaplan-Meier analysis, and the prognosis was in the following order: tubular-type, cribriform-type, and poorly differentiated-type in patients with node-positive stage IV CRC. Kaplan-Meier analysis of CSS showed that the cribriform-type group had a significantly shorter CSS (p = 0.0003) than the tubular-type group, and that the poorly differentiated-type group had a significantly shorter CSS (p = 0.0006) than the cribriform-type group (Fig. 2).

Figure 2 Kaplan–Meier plots of cancer-specific survival (CSS) in node-positive disease stage IV patients.

CSS when all metastatic lymph nodes (mLNs) showed tubular carcinoma (Tubular-type group), some mLNs showed cribriform carcinoma (Cribriform-type group), and some mLNs showed poorly differentiated carcinoma (Poorly differentiated-type group).

The association between clinicopathological factors and CSS times in patients with node-positive stage IV CRC

A Cox proportional hazards model was employed to explore factors associated with CSS in patients with node-positive stage IV disease. All patients were followed up for 5 years. There are no missing data or outliers. No adjustments were made to the Cox proportional hazards model. Before the Cox proportional hazards model, the proportional hazard assumption, Schoenfeld residuals, were tested for each factor, including sex (male vs. female) (p = 0.7901), age (>70 vs. ≤70) (p = 0.3568), tumor site (rectum vs. colon) (p = 0.1998), primary tumor differentiation (poor vs. others) (p = 0.9920), tumor depth (T4 vs. T3)(p = 0.2618), lymphatic permeation (presence) (p = 0.6739), venous permeation (presence) (p = 0.2287), number of mLNs (>3 vs. ≤3) (p = 0.0572), synchronous metastasis (vs. metachronous) (p = 0.1407), tubular-type in mLNs (vs. other types in mLNs) (p = 0.7331), and poorly differentiated-type in mLNs (vs. other types in mLNs) (p = 0.8719). Because the p-value for the cribriform-type in mLNs (vs. other types in mLNs) was 0.0418, the cribriform-type in mLNs (vs. other types in mLNs) was not analyzed in the Cox proportional hazards model. Age >70 years (hazard ratio (HR): 1.669), poorly differentiated adenocarcinoma in the primary tumor (HR: 2.352), T4 stage (HR: 1.810), lymphatic permeation (HR: 2.340), and some mLNs showing poorly differentiated carcinoma (HR: 3.554) were significantly associated with shorter CSS. The tumor site (HR: 0.524) and all mLNs showing tubular carcinoma (HR: 0.196) were significantly associated with longer CSS (Table 2).

Table 2 Univariate analyses for cancer-specific survival of patients with node-positive stage IV colorectal.

Variables	Univariate analysis	
HR	95% CI	p-value	
Gender (male vs. female)	1.229	[0.758–1.994]	0.4023	
Age (>70 vs. ≦70)	1.669	[1.009–2.762]	0.0461	
Tumor site (rectum vs. colon)	0.524	[0.316–0.867]	0.0120	
Primary tumor differentiation (por vs. others)	2.352	[1.068–5.178]	0.0337	
Tumor depth (T4 vs. T3)	1.810	[1.100–2.976]	0.0195	
Lymphatic permeation (presence)	2.340	[1.0083–5.4330]	0.0478	
Venous permeation (presence)	1.186	[0.713–1.973]	0.5101	
Number of mLNs (>3 vs. ≦3)	1.190	[0.731–1.937]	0.4842	
Synchronous metastasis (vs. metachronous)	1.421	[0.875–2.308]	0.1561	
Tubular-type in mLNs (vs. other types in mLNs)	0.196	[0.078–0.492]	0.0005	
Poorly differentiated-type in mLNs (vs. other types in mLNs)	3.554	[1.955–6.463]	<0.0001	
Note:

HR, hazard ratio; CI, confidence interval; mLN, metastatic lymph node; Tubular-type in mLNs, all mLNs showing tubular carcinoma; Cribriform type in mLNs, some mLNs showing cribriform carcinoma; Poorly differentiated-type in mLNs, some mLNs showing poorly differentiated carcinoma.

All mLNs showing tubular-type carcinoma predict better CSS in patients with node-positive stage IV CRC

Multivariate analyses of six parameters with significant differences in the univariate analysis, including age, tumor site, primary tumor differentiation, tumor depth, lymphatic permeation, and all mLNs showing tubular carcinoma, revealed that age, lymphatic permeation, and all mLNs showing tubular carcinoma were independent prognostic factors for CSS (Table 3). The HR for age, lymphatic permeation, and all mLNs with tubular carcinoma were 1.798, 2.711, and 0.196, respectively. Therefore, only “all mLNs showing tubular carcinoma” was an independent “better prognostic factor” in patients with node-positive stage IV CRC.

Table 3 Multivariate analyses, including tubular-type in mLNs, for cancer-specific survival of patients with node-positive stage IV colorectal cancer.

Variable	Multivariate analysis	
HR	95% CI	p-value	
Age (>70 vs. ≦70)	1.798	[1.074–3.011]	0.0257	
Tumor site (rectum vs. colon)	0.652	[0.381–1.114]	0.1179	
Primary tumor differentiation (por vs. others)	1.445	[0.629–3.322]	0.3856	
Tumor depth (T4 vs. T3)	1.281	[0.753–2.178]	0.3601	
Lymphatic permeation (presence)	2.711	[1.134–6.481]	0.0250	
Tubular-type in mLNs (vs. other types in mLNs)	0.196	[0.077–0.502]	0.0007	
Note:

HR, hazard ratio; CI, confidence interval; mLN, metastatic lymph node; Tubular-type in mLNs, all mLNs showing tubular carcinoma.

Some mLNs showing poorly differentiated-type carcinoma predict poor CSS in patients with node-positive stage IV CRC

Multivariate analyses of six parameters with significant differences in the univariate analysis, including age, tumor site, primary tumor differentiation, tumor depth, lymphatic permeation, and some mLNs showing poorly differentiated carcinoma, showed that age and some mLNs showing poorly differentiated carcinoma were independent prognostic factors for CSS (Table 4). The HR for age and some mLNs with poorly differentiated carcinoma were 1.806 and 2.525, respectively.

Table 4 Multivariate analyses, including poorly differentiated-type in mLNs, for cancer-specific survival of patients with node-positive stage IV colorectal cancer.

Variable	Multivariate analysis	
HR	95% CI	p-value	
Age (70> vs. ≦70)	1.806	[1.077–3.029]	0.0251	
Tumor site (rectum vs. colon)	0.680	[0.394–1.172]	0.1651	
Primary tumor differentiation (por vs. others)	1.048	[0.409–2.686]	0.9229	
Tumor depth (T4 vs. T3)	1.432	[0.829–2.476]	0.1980	
Lymphatic permeation (presence)	1.988	[0.838–4.714]	0.1187	
Poorly differentiated-type in mLNs (vs. other types in mLNs)	2.525	[1.209–5.271]	0.0136	
Note:

HR, hazard ratio; CI, confidence interval; mLN, metastatic lymph node; Poorly differentiated-type in mLNs, some mLNs showing poorly differentiated carcinoma.

Discussion

Establishing an accurate indicator of the prognosis of patients with stage IV CRC will enhance the selection of appropriate treatments for CRC. To improve the prognosis of stage IV CRC, it is necessary to first extract a better prognosis group based on current treatment, including surgery, chemotherapy, and radiotherapy, so that patients can choose the best treatment from the currently available treatment option for longer survival. However, it is necessary to consider treatment policies for the poor prognosis group, and novel treatments for this group need to be developed. In the current study, tubular-type carcinomas in the mLN had a better prognosis. In contrast, poorly differentiated carcinoma in the mLN had a poor prognosis. The tubular-type group may have a long-term prognosis with current chemotherapy and surgery. In contrast, the poorly differentiated-type group is not likely to achieve long-term survival with current therapies. Therefore, changes in current chemotherapy regimens, indications for immune checkpoint inhibitors, introduction of chemotherapy for other cancers, or development of novel anticancer agents may be needed to improve the prognosis of the poorly differentiated-type group. Although further studies are required, our classification system may guide treatment decisions for patients with different prognoses.

In the good prognosis group with stage IV CRC, the goal was to achieve complete long-term regression. Recently, the use of chemotherapy in patients with stage IV CRC has progressed significantly. Some cases achieve complete response with multidisciplinary therapy including chemotherapy (Aomatsu et al., 2021; Arata et al., 2016; Baik et al., 2021; Baimas-George et al., 2018; Bulajic et al., 2019; Dang et al., 2020; Higgins et al., 2021; Lara-Morga et al., 2023; Li et al., 2021; Tokuhara et al., 2019; Yoshida et al., 2017). Currently, we do not fully understand which cases lead to a complete response. If a group with a high possibility of complete remission can be selected, a more effective treatment can be provided. In this study, of the 18 patients with tubular-type carcinoma of the mLN, 11 had recurrence-free survival for 5 years. Current treatments, including surgery and chemotherapy, for the tubular-type group may lead to complete regression. Our classification, based on the shape of the hollow and cancer cell polarity in the mLN, has the potential to provide a better understanding of cancer malignancy and therapeutic susceptibility.

A prognostic prediction has been performed using LNM as an indicator of stage IV CRC. Positive LNM (Ishihara et al., 2014; Kuo et al., 2021), the number of positive lymph nodes (Han et al., 2018, 2020; Li et al., 2017), and the ratio of the number of positive lymph nodes to the total number of retrieved lymph nodes (lymph node ratio) (Ahmed et al., 2016; Fu et al., 2015; Jiang et al., 2019; Ozawa et al., 2015) have been reported as prognostic indicators. However, these are not sufficient to classify the malignant phenotypes of individual cases; thus, more accurate indicators are required. Cancer histology in the mLN using conventional pathological classification has been performed for stage III CRC but not for stage IV CRC (Hirayama et al., 2018; Takahashi, Mori & Yasuno, 2000). In this study, we investigated the prognosis of patients with stage IV CRC using a classification system based on the shape of the hollow cavity and cancer cell polarity. “All mLNs showing tubular carcinoma” was an independent good prognostic factor, and “some mLNs showing poorly differentiated carcinoma” was an independent poor prognostic factor for CSS in patients with node-positive stage IV CRC. Our classification of cancer histology in the mLN may distinguish between patients with stage IV CRC with good or poor prognosis and allow individualized treatment plans.

In the present study, the survival rate was different between the tubular-type and poorly differentiated-type groups. Cancer cell biology should be considered in further interpretation. LNM is the metastatic front. Therefore, LNM histology may indicate characteristics of metastatic cancer cells. Our classification, based on cancer cell polarity, mainly refers to epithelial or mesenchymal cell types. Poorly differentiated-type histology in mLN may be more mesenchymal “in the metastasis front” than tubular-type histology, resulting in poor prognosis. Moreover, in a clinical setting, poorly differentiated carcinomas in the mLN may be resistant to current chemotherapy. Further investigation of the cellular mechanism in LNM is required to address the malignant properties of poorly differentiated carcinomas in mLN.

We have previously reported an association between cribriform-type carcinoma in LNM and poor survival (Yokoyama et al., 2021, 2023). In this study, cribriform-type mLNs (vs. other types in mLNs) were not analyzed in the Cox proportional hazards model because the p-value for cribriform-type mLNs (vs. other types in mLNs) was 0.0418. In the current study, the percentages of patients with all tubular-type, some cribriform-type carcinoma, and some poorly differentiated-type carcinoma in LNM in stage IV CRC were 18.4%, 64.4%, and 17.2%, respectively. Other types in mLNs include tubular and poorly differentiated-types in mLNs. Therefore, during the first half of the observation period, the poorly differentiated-type group showed more reduced numbers than the cribriform-type group. In contrast, during the latter half, the tubular-type group maintained the number better than the cribriform-type group (Fig. 2). This may explain why the proportional assumption was not met in cribriform-type in mLNs (vs. other types in mLNs).

Limitations of the present study include the small number of patients and its retrospective design. To address the significance of mLN histology in stage IV CRC, further investigations, including retrospective studies with large sample sizes, long-term follow-up, and prospective clinical trials, are required.

Conclusions

This study demonstrated that tubular-type cancer histology in LNM from CRC indicates better prognostic properties. In contrast, the poorly differentiated-type in LNM suggests a poor prognosis in patients with node-positive stage IV CRC. Further examination of cancer histology in LNM in node-positive stage IV CRC might provide clues for appropriate treatment of stage IV CRC.

Supplemental Information

Supplemental Information 1 Raw data of patients’ clinicopathological factors.

We would like to thank Editage for English language editing.

Additional Information and Declarations

Competing Interests

Author Contributions

Human Ethics

Data Availability

The authors declare that they have no competing interests.

Shozo Yokoyama conceived and designed the experiments, performed the experiments, analyzed the data, prepared figures and/or tables, authored or reviewed drafts of the article, and approved the final draft.

Takashi Watanabe performed the experiments, analyzed the data, prepared figures and/or tables, authored or reviewed drafts of the article, and approved the final draft.

Shuichi Matsumura performed the experiments, analyzed the data, prepared figures and/or tables, authored or reviewed drafts of the article, and approved the final draft.

Masato Tamiya performed the experiments, authored or reviewed drafts of the article, and approved the final draft.

Shotaro Nagano performed the experiments, authored or reviewed drafts of the article, and approved the final draft.

Yuya Hori performed the experiments, authored or reviewed drafts of the article, and approved the final draft.

The following information was supplied relating to ethical approvals (i.e., approving body and any reference numbers):

The ethics committee of the National Hospital Organization Minami Wakayama Medical Center approved the study (#2023-11).

The following information was supplied regarding data availability:

The raw measurements are available in the Supplemental File.

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
