# Peer review of "Cancer histology in metastatic lymph node predicts prognosis in patients with node-positive stage IV colorectal cancer"

_PeerJ, doi:10.7717/peerj.17702_

## Round 0.1 · original submission · Minor Revisions

Based on the reviewers, a minor revision is needed

Reviewer 1 ·

Basic reporting

1. Clarity and language
The manuscript is written okay, using clear, professional English throughout. However, there are a number of areas where the language could be further improved to increase clarity. Specific examples include:
Line 17: “Stage IV colorectal cancer (CRC) patients need an appropriate indicator for prognosis, because treatment plan is determined by the prognosis. “ This sentence can be rephrased to improve flow and readability.
Line 42: “If the good prognosis group and the poor prognosis group can be selected, more effective treatment can be performed.” Consider rephrasing for clarity. clarity.
There are other good similarities, so please revise the entire text completely. Consider having another fluent English speaker or professional editing service review the manuscript.

2. Structure, graphics, and data sharing
The description of the “Methods of Histological Analysis” in the article has a certain level of detail, but could be further improved to increase reproducibility and clarity. This includes describing detailed staining procedures such as staining time, concentration, temperature, etc., and describing how different types of features are identified and categorized.

Experimental design

1. relevance to the scope of the journal: the study fits well within the scope of PeerJ and addresses an important issue in the field of oncology and pathology.

2. methodological details:
Provide more details about the Kaplan-Meier and Cox proportional risk models used, e.g., describe how cases with identical survival times are treated, e.g., whether adjusted Kaplan-Meier curves are used; illustrate how the Cox proportional risk model handles missing data and outliers.

3 Implications and novelty
The content and conclusions of this paper are not considered novel, but the good thing is that the data, methods and conclusions are SOLID.

Validity of the findings

1. Strengths and advantages
Large sample size and long-term follow-up to obtain data

Additional comments

This paper is relatively simple in content, data and analysis, but overall it seems to be fine. The conclusions also make sense, with a large amount of data supporting the different prognostic outcomes for different histologic types.

Reviewer 2 ·

Basic reporting

1. The writing in this paper needs to be carefully improved.
2. In table 1, it should be ‘gender’, not ‘gander’.

Experimental design

1. For the use of Cox proportional hazards model, the proportional hazards assumption needs to be tested (such as evaluating Schoenfeld residuals to examine validity of the proportional hazards assumption).
2. A table 1 with distribution of baseline characteristics (eg demographics, health conditions) should be provided, along with providing a description of the loss to follow up rate (for the 5-year follow-up mentioned).

Validity of the findings

No comment.

---

## Round 0.2 · accepted · Accept

The minor corrections have been addressed and the manuscript is accepted.